# Gonococcal OMV-delivered PorB induces epithelial cell mitophagy

Shuai Gao[1,5], Lingyu Gao[1,5], Dailin Yuan[1,2], Xu'ai Lin[1] & Stijn van der Veen [1,3,4] ✉

The bacterial pathogen *Neisseria gonorrhoeae* is able to invade epithelial cells and survive intracellularly. During this process, it secretes outer membrane vesicles (OMVs), however, the mechanistic details for interactions between gonococcal OMVs and epithelial cells and their impact on intracellular survival are currently not established. Here, we show that gonococcal OMVs induce epithelial cell mitophagy to reduce mitochondrial secretion of reactive oxygen species (ROS) and enhance intracellular survival. We demonstrate that OMVs deliver PorB to mitochondria to dissipate the mitochondrial membrane potential, resulting in mitophagy induction through a conventional PINK1 and OPTN/NDP52 mechanism. Furthermore, PorB directly recruits the E3 ubiquitin ligase RNF213, which decorates PorB lysine residue 171 with K63-linked poly-ubiquitin to induce mitophagy in a p62-dependent manner. These results demonstrate a mechanism in which polyubiquitination of a bacterial virulence factor that targets mitochondria directs mitophagy processes to this organelle to prevent its secretion of deleterious ROS.

*N eisseria gonorrhoeae* is a human-specific bacterial pathogen that colonizes the mucosal sites in the urogenital tract, pharynx, and rectum to induce the sexual transmitted disease gonorrhoea. *N. gonorrhoeae* is able to penetrate the mucosal epithelia where it modulates epithelial shedding[1–3]. It is furthermore able to invade epithelial cells to transmigrate to subepithelial spaces, which may result in disseminated gonococcal infections[4–6]. Intracellular *N. gonorrhoeae* is targeted by autophagy and subsequently delivered to lysosomes for destruction, with only a subpopulation of intracellular bacteria escaping this process[7–9]. During infection and colonization of mucosal epithelia, *N. gonorrhoeae* releases outer membrane vesicles (OMVs) that trigger initiation of innate immune responses[10].

Secretion of OMVs provides a mechanism for pathogenic bacteria to modulate innate and adaptive immune responses, or modulate the host environment[11,12]. OMVs are detected by NOD1 in epithelial cells[13,14], which can result in activation of autophagy responses[13,15,16] or prime autophagy for subsequent bacterial invasion[17]. Gonococcal OMVs are able to deliver the outer membrane porin protein PorB to macrophage mitochondria to induce apoptosis[18,19], while ectopic expression of PorB in epithelial cells induces a pre-apoptotic state[20]. Damaged or dysfunctional mitochondria are degraded by a mitochondria-selective autophagy process termed mitophagy, which prevents induction of apoptosis upon mitochondrial damage[21–23]. It is currently unknown whether gonococcal OMVs impact intracellular survival in epithelial cells and how OMVs interact with selective autophagy processes.

Here, we show that gonococcal OMVs induce mitophagy in epithelial cells though a dual PorB-dependent mechanism. OMVs are endocytosed by epithelial cells and delivered to mitochondria for subsequent translocation of PorB. PorB dissipates the mitochondrial membrane potential (MMP) to induce PINK1-dependent mitophagy. A second mitophagy mechanisms results from PorB decoration with K63-linked polyubiquitin by the E3 ubiquitin ligase RNF213. Induction of mitophagy reduces mitochondrial secretion of reactive oxygen species (ROS), which enhances gonococcal intracellular survival.

[1]Department of Microbiology, and Department of Dermatology of Sir Run Run Shaw Hospital, School of Medicine, Zhejiang University, Hangzhou, PR China. [2]Zhejiang University-University of Edinburgh Institute, Zhejiang University, Haining, PR China. [3]State Key Laboratory for Diagnosis and Treatment of Infectious Diseases, Collaborative Innovation Center for Diagnosis and Treatment of Infectious Diseases, The First Affiliated Hospital, School of Medicine, Zhejiang University, Hangzhou, PR China. [4]Zhejiang Provincial Key Laboratory for Microbial Biochemistry and Metabolic Engineering, Hangzhou, PR China. [5]These authors contributed equally: Shuai Gao, Lingyu Gao. ✉e-mail: stijnvanderveen@zju.edu.cn

## Results

### Gonococcal OMVs disrupt epithelial cell mitochondria and activate mitophagy

*N. gonorrhoeae* abundantly secretes OMVs during invasion and exocytosis of epithelial cells (Fig. 1a). It has previously been shown that deletion of *vacJ* from the bacterial genome results in increased OMV secretion[24,25]. Therefore, we generated a gonococcal Δ*vacJ* deletion mutant of the international reference strain ATCC 49226, which indeed showed increased secretion of OMVs (Supplementary Fig. 1a). Importantly, the Δ*vacJ* deletion mutant showed increased intracellular

survival and exocytosis of epithelial cells (Fig. 1b) and decreased colocalization with lysosomal marker LAMP1 (Fig. 1c). Interestingly, the Δ*vacJ* mutant showed increased activation of autophagy marker protein LC3 and accumulation of LC3 puncta compared with the wild-type strain (Supplementary Fig. 1b, c), but decreased colocalization with LC3 puncta (Supplementary Fig. 1c), pointing to a role for OMVs in this process. Gonococcal OMVs are endocytosed by epithelial cells in a dynamin-dependent manner (Supplementary Fig. 2), and purified OMVs from both strain ATCC 49226 and contemporary clinical isolate ZJXSH86 (Fig. 1d) subsequently activated autophagy fluxes (Fig. 1e, f).

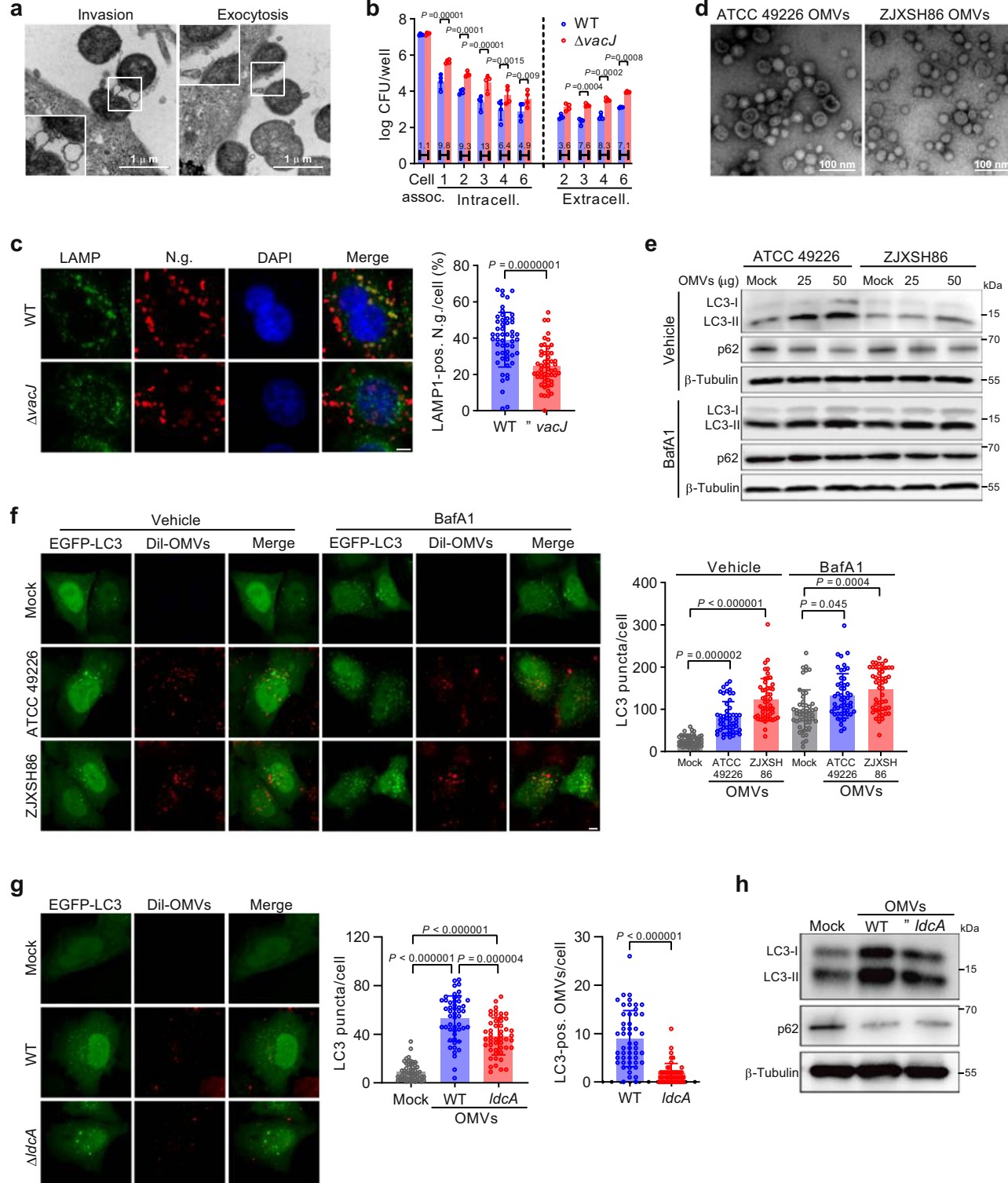

**Fig. 1 | Gonococcal OMVs induce autophagy flux in epithelial cells. a** TEM of gonococcal OMV secretion during invasion and exocytosis of HeLa cells. Images are representative of 2 independent experiments. **b** Enhanced intracellular survival and exocytosis of gonococcal Δ*vacJ* mutant in gentamicin protection assays with HeLa cells. Data are mean ± s.d. of log-normalized colony forming units (CFU) per well and relative differences in survival between WT and Δ*vacJ* are provided within the bars; *n* = 4 independent experiments, two-way ANOVA with posthoc Bonferroni test. **c** Δ*vacJ* mutant shows reduced LAMP1-colocalization after a one-hour challenge of HeLa cells. Scale bar, 5 μm. Data are mean ± s.d.; *n* = 53 cells, unpaired two-tailed *t*-test. **d** TEM images of purified gonococcal OMVs. Images are representative of 4 (ATCC 49226) and 3 (ZJXSH86) independent experiments. **e** LC3 Western blots show induction of autophagy flux in HeLa cells stimulated with gonococcal OMVs

from strains ATCC 49226 and ZJXSH86. **f** Gonococcal OMVs induce accumulation of LC3 puncta in HeLa cells. Scale bar, 5 μm. Data are mean ± s.d.; *n* = 50 cells, Kruskal–Wallis with posthoc Dunn test (Vehicle Mock-ZJXSH86 OMVs: *P* < 10⁻¹⁵). **g** OMVs from the gonococcal Δ*ldcA* mutant induce accumulation of LC3 puncta in HeLa cells, but largely lost colocalization with LC3 puncta. Scale bar, 5 μm. Data are mean ± s.d.; *n* = 50 cells, one-way ANOVA with post-hoc Tukey test for quantification of LC3 puncta (Mock-WT OMVs: *P* = 3 × 10⁻¹⁴; Mock-Δ*ldcA* OMVs: *P* = 6 × 10⁻¹⁴), unpaired two-tailed *t*-test for quantification of LC3-positive OMVs (*P* = 5 × 10⁻¹³). **h** LC3 Western blots of OMV-stimulated HeLa cells show reduced accumulation of LC3 for the Δ*ldcA* mutant. Cells in **c**, **f**, **g** are from 3 independent experiments. Western blots in **e**, **h** are representative of 3 independent experiments. Source data are provided as a Source Data file.

Human NOD1 detects bacterial GlcNAc-MurNAc tripeptide motif (GM-Tri_DAP)-containing peptidoglycan fragments[26], while a gonococcal Δ*ldcA* mutant is unable to generate and secrete GM-Tri_DAP peptidoglycan and therefore escapes NOD1 sensing[9,27] and NOD1-dependent autophagy targeting[9,15,16]. Gonococcal OMVs obtained from the Δ*ldcA* mutant still activated autophagy, albeit at lower levels then OMVs from the WT strain (Fig. 1g, h), but OMVs from the Δ*ldcA* mutant largely escaped direct autophagy targeting based on minimal colocalization with LC3 puncta (Fig. 1g). Therefore, it appears that besides a direct NOD1-dependent pathway, gonococcal OMVs also activate autophagy through an alternative mechanism.

Gonococcal OMVs did not display epithelial cell cytotoxicity or induce apoptosis (Supplementary Fig. 3). However, endocytosis of gonococcal OMVs resulted in the specific degradation of mitochondrial marker proteins TOM20 and TIM23, but not of GM130 (Golgi) or PDI (endoplasmic reticulum) (Fig. 2a), indicating that OMVs might induce mitophagy. Indeed, OMVs surrounding or engaging with mitochondria were readily observable (Fig. 2b), while high-resolution confocal microscopy with the endosome tracker dextran cascade blue showed mitochondria surrounded by OMVs that are predominantly present within endosomes (Fig. 2c). Furthermore, the mitochondrial membrane potential (MMP) dissipated after epithelial cell endocytosis of OMVs (Fig. 2d), resulting in mitochondrial disruption and capture by autophagosomal structures and lysosomes (Fig. 2e), for subsequent degradation (Fig. 2f). Further confocal microscopy analysis showed strong colocalization between mitochondria (HSP60) and lysosomes (LAMP1) (Fig. 2g) and mitochondria and LC3 (Fig. 2h), confirming the role of OMVs in the induction of mitophagy. Similarly, epithelial cell invasion by the Δ*vacJ* mutant resulted in increased disruption of mitochondria and degradation of mitochondrial marker proteins (Supplementary Fig. 1d, e).

## OMV-delivered PorB dissipates MMP and induces PINK1- and OPTN/NDP52-dependent mitophagy

It has previously been shown that ectopic expression of gonococcal PorB, but not PorB from the closely related *Neisseria mucosa*, dissipates MMP in epithelial cells[20]. Indeed, ectopically expressed PorB colocalized with mitochondria to dissipate MMP (Supplementary Fig. 4a, b), which resulted in mitochondrial disruption and capture in autophagosomal structures (Fig. 3a), while PorB of *N. mucosa* is unable to localize at the mitochondria or dissipate MMP (Supplementary Fig. 4a, b). Gonococcal PorB is a polymorphic β-barrel divided into two main types, PorB1a and PorB1b, which are expressed from the same genetic locus and share approximately 60–80% amino acid sequence identity[28]. Ectopic expression of gonococcal PorB1a from strain ATCC49226 and PorB1b from strain ZJXSH86, resulted in reduced levels of mitochondrial marker proteins and induction of LC3 expression, while PorB from *N. mucosa* did not affect the levels of these marker proteins (Fig. 3b). PorB is an essential gonococcal protein and can therefore not be deleted. However, it was possible to replace gonococcal *porB* with *porB* from *N. mucosa* using a homologous

recombination method that also places a downstream kanamycin resistance cassette. Gonococcal OMVs containing PorB from *N. mucosa* were unable to dissipate MMP or induce mitophagy (Fig. 3c–e), indicating that PorB is the only protein required for OMV-mediated induction of mitophagy. PorB-mediated MMP dissipation is dependent on ATP binding via lysines within the PorB channel[20,29]. Specific replacement of lysine 117 with glutamine (Fig. 3f) largely abolishes ATP binding and limits opening of the PorB channel[20]. Ectopically-expressed PorB K117Q still localized at mitochondria (Supplementary Fig. 4c), but indeed did not dissipate MMP (Supplementary Fig. 4d). Importantly, the PorB K117Q mutant showed reduced degradation of mitochondrial marker proteins (Fig. 3g) and reduced colocalization between HSP60 and LC3 or LAMP1 (Fig. 3h, i), implying reduced mitophagy compared with the wild-type PorB, although residual mitophagy activity was still detected.

LC3-coated autophagosomal membranes recognize target substrates that are decorated with autophagy receptor proteins[30–33]. Wild-type PorB is unable to induce mitophagy in a HeLa quadruple knock-out cell line (4KO) for the autophagy receptor proteins OPTN, NDP52, p62 and NBR1, while mitophagy is rescued by transfection with expression vectors for OPTN, NDP52 or p62 (Fig. 3j). In contrast, mitophagy induced by the PorB K117Q mutant is only rescued by p62 in this cell line (Fig. 3j). HeLa cells do not express Parkin and therefore conventional MMP-dissipation-dependent mitophagy is dependent on PINK1 and facilitated by OPTN and NDP52[34]. The receptors OPTN, NDP52, and p62 are commonly associated with lysine 63-linked poly-ubiquitin loaded substrates[31,35,36]. Therefore, we generated a PorB mutant in which all 20 lysines were replaced with glutamines (PorB K_null) and a PorB mutant only retaining the five lysines in the PorB channel (PorB K5) to assure ATP binding and dissipation of MMP (Fig. 3f). Both PorB K5 and PorB K_null retained the ability to target mitochondria, but only PorB K_null lost the ability to dissipate MMP (Supplementary Fig. 4e, f). Furthermore, the PorB K_null mutant was no longer able to induce mitophagy, but interestingly, the PorB K5 mutant showed similar residual mitophagy activity as the PorB K117Q mutant (Fig. 3k). The PorB K5 mutant was still rescued for PorB-induced mitophagy in the HeLa 4KO cell line transfected with OPTN or NDP52 expression vectors, but not by p62 expression (Fig. 3j). Silencing of PINK1 in wild-type HeLa cells inhibited PorB K5-induced mitophagy, but not PorB K117Q-induced mitophagy (Fig. 3l), indicating that PorB-induced MMP-dissipation-dependent mitophagy follows the conventional PINK1- and OPTN/NDP52-dependent pathway in HeLa cells.

## K63-linked polyubiquitination of PorB by RNF213 induces PorB- and p62-dependent mitophagy

The role of p62 in PorB-induced mitophagy in a mechanism independent of MMP dissipation was confirmed by silencing of p62, which inhibits PorB K117Q-induced mitophagy (Fig. 4a). Furthermore, wild-type PorB and PorB K117Q induced strong p62 colocalization with mitochondria, but not the PorB K5 or PorB K_null mutants (Fig. 4b). Induction of mitophagy required both the p62 LIR domain,

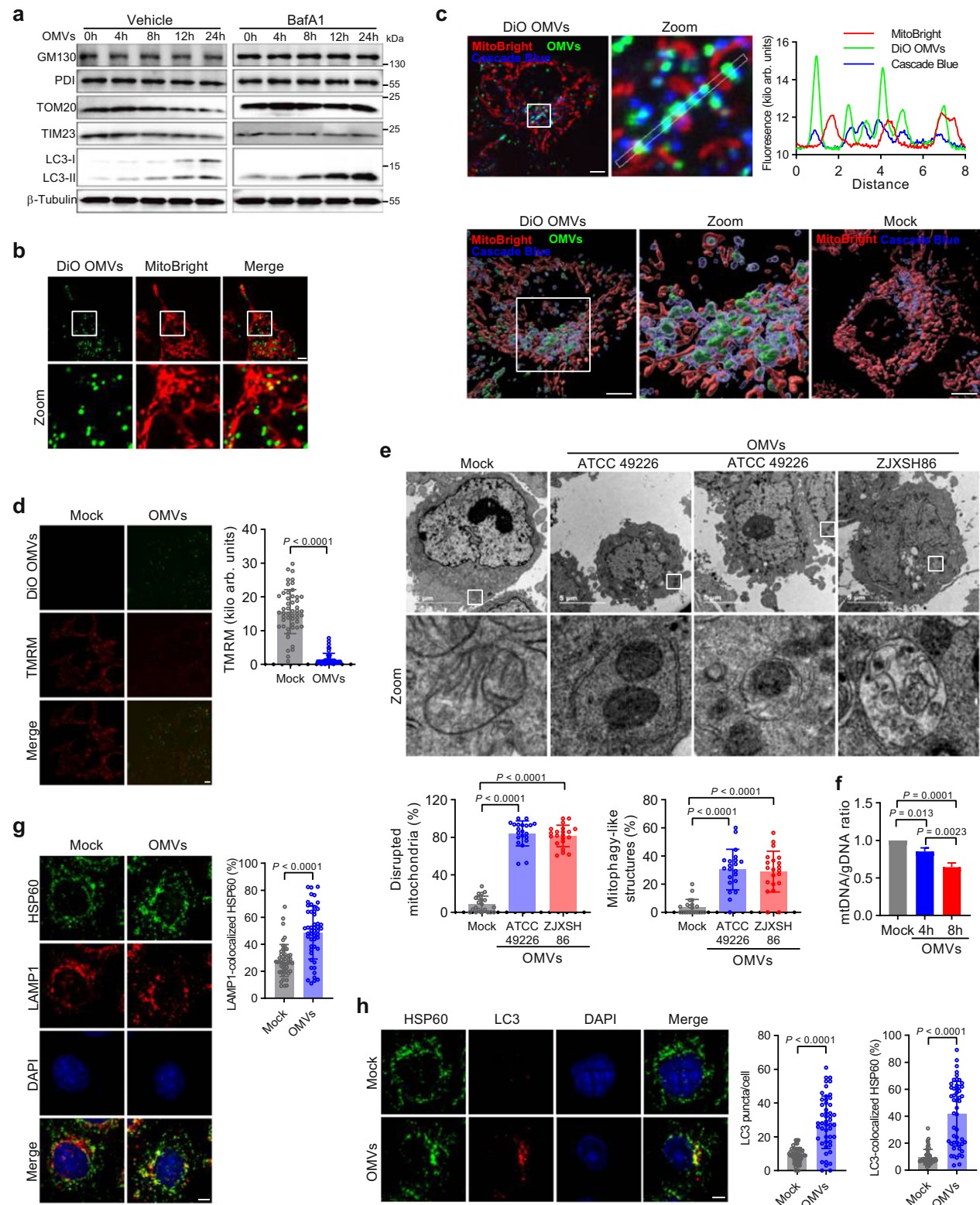

which is responsible for LC3 interactions, and the p62 ubiquitin binding domain UBA, but its PB1 domain, responsible for p62 oligomerization and proteasomal targeting, was dispensable for p62-dependent mitophagy induced by PorB K117Q (Fig. 4c). Subsequent immunoprecipitation of PorB from epithelial cells transfected with an expression vector for ubiquitin resulted in co-immunoprecipitation of both polyubiquitin and p62 for wild-type PorB and PorB K117Q (Fig. 4d). Further immunoprecipitation assays

using cells transfected with expression vectors for ubiquitin that can only form K63-linked or K48-linked polyubiquitin showed that PorB is decorated with both K63- and K48-linked polyubiquitin, resulting in K63 polyubiquitin-dependent autolysosomal degradation and K48 polyubiquitin-dependent proteasomal degradation of PorB (Fig. 4e). However, only K63-linked polyubiquitin showed strong mitochondrial colocalization (Fig. 4f). Screening of specific PorB lysine residues showed that lysine 171 is decorated with K63-linked

**Fig. 2 | Gonococcal OMVs target mitochondria and induce mitophagy.**
**a** Western blots of OMV-stimulated HeLa cells show specific autophagosome/
lysosome-dependent degradation of mitochondrial marker proteins (TOM20 and
TIM23), but not of Golgi (GM130) or endoplasmic reticulum (PDI). Western blots
are representative of 3 independent experiments. **b** DiO-labeled gonococcal OMVs
from strain ATCC 49226 colocalize with MitoBright-labeled mitochondria in HeLa
cells. Scale bar, 5 μm. **c** Live HeLa cell microscopy and 3D image reconstruction
shows OMVs remain associated with Cascade Blue-labeled endosomes when
delivered to MitoBright-labeled mitochondria. Scale bar, 5 μm. The fluorescence
colocalization profile of the line is shown. **d** Gonococcal OMVs from strain ATCC
49226 dissipate the mitochondrial membrane potential (MMP) in HeLa cells. Scale
bar, 5 μm. Data are mean ± s.d.; $n = 50$ cells, unpaired two-tailed $t$-test, $P < 10^{-15}$.
**e** TEM of HeLa cells stimulated with gonococcal OMVs from strains ATCC 49226
and ZJXSH86 show mitochondrial disruption and capture in mitophagy-like
structures. Data are mean ± s.d.; $n = 21$ cells, one-way ANOVA with post-hoc Tukey

test for mitochondrial disruption (Mock-ATCC 49226 OMVs: $P = 2 \times 10^{-11}$; Mock-
ZJXSH86 OMVs: $P = 2 \times 10^{-11}$), Kruskal–Wallis with posthoc Dunn test for mito-
chondria in mitophagy-like structures (Mock-ATCC 49226 OMVs: $P = 1 \times 10^{-6}$; Mock-
ZJXSH86 OMVs: $P = 7 \times 10^{-6}$). **f** Quantitative real-time PCR showing a reduced
mitochondrial to genomic DNA ratio in ATCC 49226 OMV-stimulated HeLa cells.
Data are mean ± s.d.; $n = 3$ independent biological replicates, one-way ANOVA with
post-hoc Tukey test. **g** Increased HSP60 and LAMP1 colocalization in ATCC 49226
OMV-stimulated HeLa cells. Scale bar, 5 μm. Data are mean ± s.d.; $n = 50$ cells,
unpaired two-tailed $t$-test, $P = 6 \times 10^{-9}$. **h** Increased HSP60 and LC3 colocalization in
ATCC 49226 OMV-stimulated HeLa cells. Scale bar, 5 μm. Data are mean ± s.d.;
$n = 50$ cells, two-tailed Mann–Whitney test (LC3 puncta Mock-OMVs: $P = 7 \times 10^{-12}$;
LC3 colocalized HSP60 Mock-ZJXSH86 OMVs: $P = 4 \times 10^{-15}$). Cells in **d, e, g, h** are
from 3 independent experiments. Images in **b, c** are representative of 3 indepen-
dent experiments. Source data are provided as a Source Data file.

polyubiquitin and lysine 128 with K48-linked polyubiquitin (Fig. 4g,
Supplementary Fig. 5). However, only lysine 171, and not lysine 128,
was involved in PorB-induced mitophagy (Fig. 4h), with mitophagy
fully inhibited for the PorB K117Q/K171Q double mutant. Similarly,
OMVs isolated from gonococcal strains expressing PorB mutants
showed that OMV-induced mitophagy was fully inhibited for the
K117Q/K171Q double mutant, while the K117Q and K171Q single
mutants showed reduced mitophagy activity compared with wild-
type PorB (Fig. 4i). PorB ubiquitination of the homologous lysine
residue 170 and subsequent induction of mitophagy was also
observed for PorB1b from strain ZJXSH86, which shares 70% amino
acid sequence identity with PorB1a from strain ATCC 49226 (Sup-
plementary Fig. 6a–c), indicating that this mechanism is not restric-
ted to the PorB1a type. Furthermore, PorB-induced mitophagy was
also observed in HCT116 and HEK293T cells (Supplementary Fig. 6d).

Polyubiquitination of target substrates is performed by the large
family of E3 ubiquitin ligases, for which approximately 650 members
have thus far been identified in human cells[37]. Immunoprecipitation
of PorB from epithelial cells and subsequent mass-spectrometry
analysis of co-immunoprecipitated proteins identified a peptide
belonging to the E3 ubiquitin ligase ring finger protein 213 (RNF213)
(Fig. 5a). Subsequent immunoprecipitation of PorB from cells trans-
fected with a FLAG-RNF213 expression vector resulted in the co-
immunoprecipitation of FLAG-RNF213 (Fig. 5b), and confocal
microscopy analysis showed strong co-localization between RNF213
and mitochondrial-localized PorB (Fig. 5c, d). RNF213 specifically
polyubiquitinates PorB lysine residue 171, since overexpression of
RNF213 enhanced K171-specific polyubiquitination (Fig. 5e) and
polyubiquitination of this lysine was abolished after silencing of
RNF213 (Fig. 5f). Similarly, overexpression or silencing of RNF213
enhanced or reduced PorB K117Q-induced mitophagy, respectively,
while mitophagy induced by PorB K171Q remained unaffected
(Fig. 5g, h), proving that RNF213 is the E3 ubiquitin ligase involved in
PorB-induced mitophagy through K63-linked polyubiquitination.

### OMV-induced mitophagy reduces mitochondrial ROS to enhance intracellular survival

Although mostly established for phagocytic cells, mitochondria are
able to detect intracellular pathogens and respond by inducing the
production of ROS to restrict intracellular survival[38–40]. Endocytosis of
gonococcal OMVs also resulted in transient induction of mitochondrial
ROS (Fig. 6a), which was prolonged when mitophagy was inhibited
with Mdivi-1 (Fig. 6b and Supplementary Fig. 7a), indicating that OMV-
induced mitophagy restricts mitochondrial ROS formation. Mitophagy
affects intracellular survival of *N. gonorrhoeae*, since pretreatment of
cells with Mdivi-1 reduced gonococcal survival, while induction of
mitophagy with CCCP or quenching of mitochondrial ROS with mito-
TEMPO enhanced intracellular survival (Fig. 6c). Similarly, pre-
exposure of epithelial cells to gonococcal wild-type OMVs enhanced

intracellular survival as a result of reduced production of mitochon-
drial ROS, which was not observed using OMVs expressing PorB from
*N. mucosa* or PorB K117Q/K171Q (Fig. 6d and Supplementary Fig. 7b, c).
In contrast, pre-exposure of Mdivi-1 pretreated cells to gonococcal
OMVs reduced intracellular survival (Fig. 1e), indicating that OMV-
induced enhanced survival is dependent on induction of mitophagy.
Finally, gonococcal invasion of epithelial cells also induced a transient
spike in mitochondrial ROS, which was prolonged for *N. gonorrhoeae*
expressing PorB from *N. mucosa* or PorB K117Q/K171Q and reduced for
the OMV-overproducing Δ*vacJ* mutant (Fig. 6f), which corresponded
with reduced or enhanced intracellular survival (Fig. 6g), highlighting
the role of OMV-induced mitophagy in intracellular survival.

## Discussion

Mitochondria are highly structured organelles with a large surface area
and therefore they play an important role as cellular sensors, including
for invading pathogens[41,42]. For phagocytic cells it has been demon-
strated that upon detection of bacterial pathogens mitochondrial ROS
production is enhanced to kill the intracellular bacteria[38,40,43–45]. To
avoid mitochondrial detection, bacterial pathogens use various
mechanisms to disrupt mitochondrial functionality, resulting in
mitochondrial fission, fusion, or apoptosis[46,47]. OMVs can function as a
bacterial secretion system for delivery of virulence factors. OMVs from
*N. gonorrhoeae* and other bacterial pathogens are able to target the
mitochondria to activate the inflammasome and induce apoptosis or
pyroptosis in phagocytic cells[18,48,49]. In contrast, for enterohemor-
rhagic *Escherichia coli* it has been shown that OMVs deliver several
virulence factors to non-phagocytic epithelial and endothelial cells to
induce cell lysis or apoptotic cell death[50]. Only recently it was
demonstrated that some bacteria specifically induce mitophagy in
order to avoid the deleterious mitochondrial ROS effects[44,45]. *Listeria
monocytogenes* induces mitophagy through its listeriolysin O (LLO),
which stimulates oligomerization of the NOD-like receptor NLRX1 for
recruitment of LC3 to the mitochondria[45]. In contrast, *Yersinia pestis*
induces mitophagy in macrophages in a conventional PINK1/Parkin-
dependent mitophagy mechanism activated by MMP loss induced by
the effector protein YopH[44]. In phagocytic cells, gonococcal OMVs
deliver PorB to the mitochondria to dissipate MMP and concomitantly
induce apoptosis[18,19].

Here we now show that gonococcal OMVs are also able to deliver
PorB to epithelial cell mitochondria, which instead results in the acti-
vation of mitophagy to clear the disrupted mitochondria. Besides a
conventional PINK1- and OPTN/NDP52-dependent mitophagy
pathway[34] that is activated as a result of PorB-induced MMP dissipa-
tion, PorB induces mitophagy through a second mechanism in which
PorB recruits the E3 ubiquitin ligase RNF213 (Fig. 7). RNF213 subse-
quently decorates PorB with K63-linked polyubiquitin for recruitment
of p62 and LC3-coated phagophores. RNF213 is the largest E3 ubiquitin
ligase discovered in humans to date, with a molecular weight of

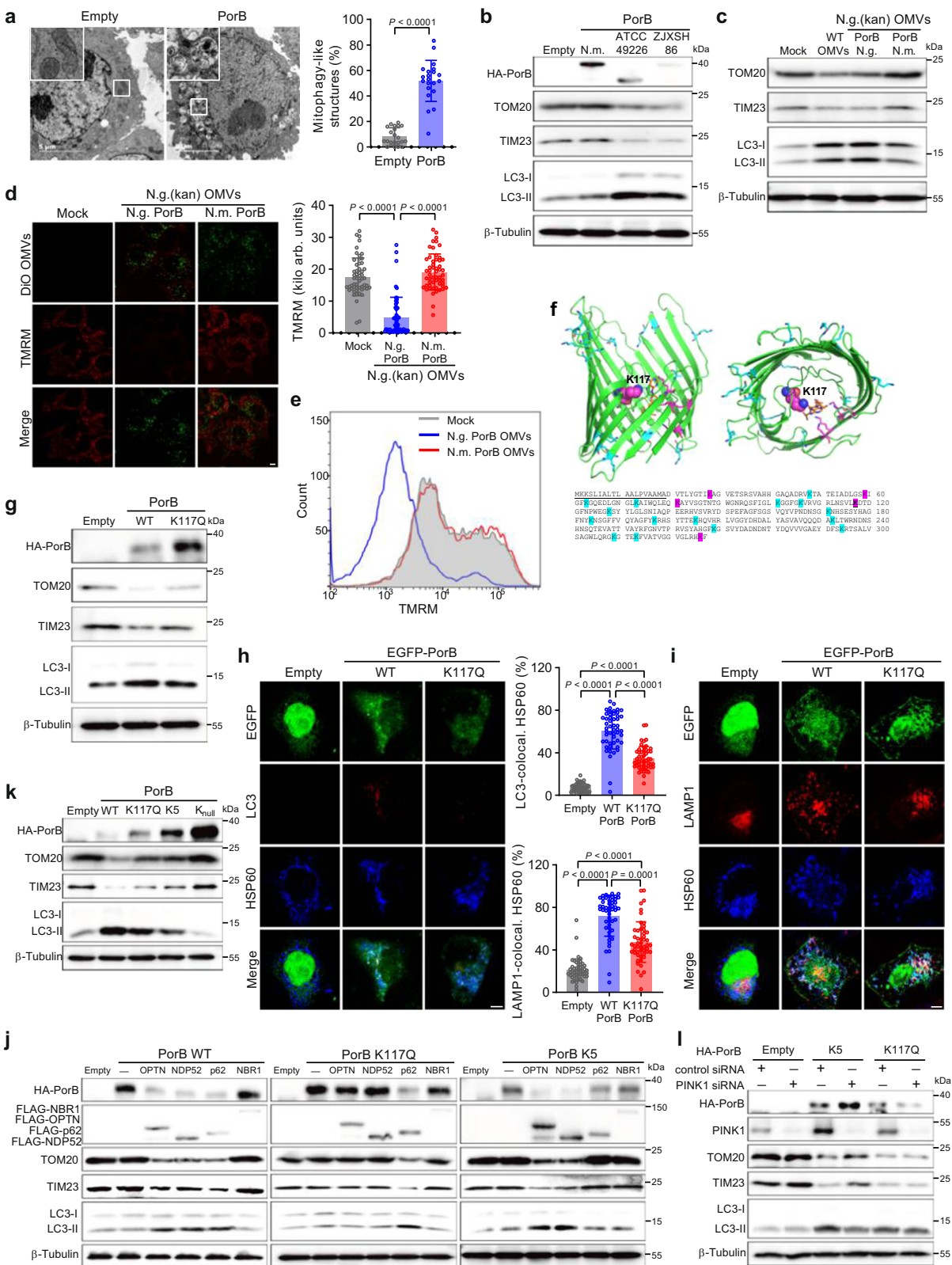

approx. 600 kD. Interestingly, while E3 ubiquitin ligases conventionally target proteins, it was recently shown that RNF213 is able to target cytosolic *Salmonella* to add polyubiquitin to the lipid A moiety of lipopolysaccharides, thereby generating a ubiquitin coat that restricts intracellular survival[51]. For *Chlamydia trachomatis* it was shown that RNF213 generates a ubiquitin coat on bacterial inclusions in IFN-γ-primed cells, which is prevented in wild-type bacteria by the

effector protein GarD[52] to escape mitochondrial sensing and deleterious effects of mitochondrial ROS.

It is currently not yet fully elucidated how gonococcal OMVs deliver PorB to the mitochondria. Our high-resolution microscopy demonstrated that the OMVs surrounding mitochondria reside abundantly within endosomes, but it is not clear whether PorB is translocated from the endosomes by direct interaction with the mitochondria

**Fig. 3 | Gonococcal OMVs deliver PorB to mitochondria to induce mitophagy.**
**a** TEM of HeLa cells expressing gonococcal PorB from strain ATCC 49226 show mitochondrial capture in mitophagy-like structures. Data are mean ± s.d.; $n = 21$ cells, two-tailed Mann–Whitney test, $P = 2 \times 10^{-10}$. **b** Western blots showing degradation of mitochondrial proteins TOM20 and TIM23 in HeLa cells expressing gonococcal PorB from strains ATCC 49226 and ZJXSH86, but not for PorB from *Neisseria mucosa*. **c** Western blots showing that gonococcal OMVs expressing PorB from *N. mucosa* lost the ability to induce degradation of TOM20 and TIM23 in HeLa cells. **d** Gonococcal OMVs expressing PorB from *N. mucosa* lost the ability to dissipate the mitochondrial membrane potential (MMP) in HeLa cells. Scale bar, 5 μm. Data are mean ± s.d.; $n = 55$ cells from 4 independent experiments, Kruskal–Wallis with posthoc Dunn test (Mock-N.g. PorB: $P = 8 \times 10^{-13}$; N.g. PorB-N.m. PorB: $P < 10^{-15}$). **e** Flow cytometry analysis of TMRM fluorescence intensity in HeLa cells stimulated with gonococcal OMVs expressing gonococcal PorB or PorB from *N. mucosa*. **f** Gonococcal PorB structure (pdb entry 4AUI) and sequence of PorB from strain ATCC 49226. Lysines are indicated in blue, or in magenta when located within the PorB channel and associated with ATP (orange) binding. **g** Western blots

showing reduced degradation of TOM20 and TIM23 in HeLa cells expressing PorB K117Q. **h** Reduced HSP60 and LC3 colocalization in HeLa cells expressing PorB K117Q compared with PorB WT. Scale bar, 5 μm. Data are mean ± s.d.; $n = 51$ cells, Kruskal–Wallis with posthoc Dunn test (Empty-WT PorB: $P < 10^{-15}$; Empty-PorB K117Q: $P = 2 \times 10^{-9}$; WT PorB-PorB K117Q: $P = 1 \times 10^{-5}$). **i** Reduced HSP60 and LAMP1 colocalization in HeLa cells expressing PorB K117Q compared with PorB WT. Scale bar, 5 μm. Data are mean ± s.d.; $n = 51$ cells, Kruskal–Wallis with posthoc Dunn test (Empty-WT PorB: $P < 10^{-15}$; Empty-PorB K117Q: $P = 9 \times 10^{-8}$). **j** Western blots of HeLa 4KO cells showing p62 expression restores PorB K117Q-dependent degradation of TOM20 and TIM23, while expression of OPTN or NDP52 restores PorB K5-dependent degradation of TOM20 and TIM23. **k** Western blots showing impaired degradation of TOM20 and TIM23 in HeLa cells expressing PorB K117Q, K5 or K$_{null}$. **l** Western blots showing that knock-down of PINK1 in HeLa cells inhibits PorB K5-dependent degradation of TOM20 and TIM23, but not for PorB K117Q. Cells in **a**, **h**, **i**, are from 3 independent experiments. Western blots in **b**, **c**, **g**, **j**, **k**, **l** are representative of 3 independent experiments. Source data are provided as a Source Data file.

---

or whether OMVs or PorB first enter the cytosol. Direct interactions between endosomes and mitochondria in a so called kiss-and-run mechanism have been demonstrated for iron transfer[53]. However, given that gonococcal PorB is polyubiquitinated by cytosol-localized RNF213, it is feasible that PorB first enters the cytosol before mitochondrial translocation. PorB is imported by the translocase of the outer mitochondrial membrane (TOM) complex and inserted in the outer membrane by the sorting and assembly machinery (SAM) with support of shuttling chaperones of the intermembrane space[20,54–56]. PorB lacks an obvious N-terminal presequence for mitochondrial targeting. Instead, its C-terminal quarter is important for mitochondrial import and sufficient to target other proteins to mitochondria[57]. Nevertheless, other PorB domains also appear to contribute to mitochondrial import. Mitochondrial β-barrel proteins are imported into mitochondria based on hydrophobicity of lipid-facing amino acids in β-hairpin motifs; particularly the exterior C-terminal β-hairpin contributes most strongly to mitochondrial import[58]. Furthermore, it appears that cytosolic chaperones contribute to their translocation and presentation to the TOM complex[59]. β-Hairpin motifs of gonococcal PorB are not as hydrophobic as the motifs from mitochondrial β-barrel proteins such as human VDAC1. Furthermore, the hydrophobicity of lipid-facing amino acids in the C-terminal quarter β-hairpins of gonococcal PorB and PorB from *N. mucosa* do not differ noteworthy. Therefore, the hydrophobicity of C-terminal β-hairpin motifs do not fully explain the inability of *N. mucosa* PorB to translocate to mitochondria. Whether interactions with cytosolic chaperones contribute to mitochondrial translocation of gonococcal PorB remains to be determined, but it is noteworthy that our PorB immunoprecipitation experiments demonstrated co-immunoprecipitation of a number of cytosolic chaperones.

In conclusion, our study highlighted a beneficial interplay between gonococcal OMVs and mitochondria. Our discovery of PorB ubiquitination by RNF213 for induction of mitophagy and enhanced intracellular survival of *N. gonorrhoeae* expands our knowledge on the repertoire by which bacterial virulence factors modulate the intracellular environment.

## Methods
### Bacterial strains
*N. gonorrhoeae* strain ATCC 49226 is an international reference strain, which is pilin-negative and Opa-positive and expresses a PorB1a-type porin protein. *N. gonorrhoeae* multidrug-resistant clinical isolate ZJXSH86[60,61] expresses a PorB1b-type porin and is pilin-positive and Opa-positive. The ATCC 49226 Δ*vacJ* mutant is a clean full-length deletion mutant that does not contain a selection marker. This mutant was constructed following a method we published previously[62,63], using primers provided in Supplementary Table 1. The ATCC 49226

Δ*ldcA* mutant was constructed following a method we published previously[64], and replaced the *ldcA* gene with the kanamycin resistance gene *kanR* using primers provided in Supplementary Table 1. For construction of *N. gonorrhoeae* ATCC 49226 strains expressing PorB mutants, the *porB* gene with 290 bp upstream region and 53 bp downstream region (including the *porB* terminator sequence), followed by the *kanR* gene and 457 bp further downstream region following the CTGTTTT repeat sequence directly downstream of the *porB* terminator sequence, were synthesized (GenScript) and cloned into vector pUC57 to generate vector pUC57-*porB*-*kanR*. The ATCC 49226 PorB control strain N.g.(kan) was generated by subsequent homologous recombination with linearized vector pUC57-*porB*-*kanR*. ATCC 49226 expressing the PorB mutants K117Q, K171Q and the double mutant K117Q/K171Q were generated by mutagenesis PCR on vector pUC57-*porB*-*kanR* using primers listed in Supplementary Table 1 and subsequent homologous recombination into the *N. gonorrhoeae* genome. To construct ATCC 49226 expressing PorB from *N. mucosa*, the complete *N. mucosa porB* open reading frame was synthesized (Tsingke Biotech) and used to replace gonococcal *porB* in vector pUC57-*porB*-*kanR* for subsequent homologous recombination into the *N. gonorrhoeae* genome. For all experiments, bacteria were revived from −80 °C glycerol stocks on GC agar (Oxoid, CM0367B) containing 1% (v/v) Vitox (Oxoid, SR0090A) and grown overnight at 37 °C in the presence of 5% $CO_2$.

### OMV purification
Overnight grown wild-type and mutant bacteria were suspended in 1 liter GC broth containing 1% Vitox and cultured at 37 °C until an $OD_{600} \approx 0.8$. Cultures were centrifuged at $12,000 \times g$ for 12 min and the supernatant was filtered (pore size 0.22 μm, Millipore, #SLGP033RB). Filtrates were centrifuged at $210,000 \times g$ for 3 h and OMV pellets were washed with PBS and finally suspended in PBS. OMV yields were quantified based on protein content with the BCA assay (Beyotime, #P0012) and OMVs were stored at −80 °C until use. When required for visualization by confocal fluorescence microscopy, OMVs were labeled with 10 μM Dil (Beyotime, C1036) or DiO (Beyotime, #C1038) for 10 min at room temperature.

### Cell culture and gentamicin protection assays
HeLa cells (ATCC, #CCL-2), HEK293T cells (ATCC, CRL-3216) and HCT116 cells (ATCC, CCL-247) were maintained in Dulbecco's Modified Eagle Medium (Gibco, #11965118) supplemented with 10% FBS (Gibco, #C0232) at 37 °C with 5% $CO_2$. The HeLa quadruple knock-out (4KO) cells for optineurin (OPTN), sequestosome 1, (SQSTM1/p62), nuclear dot protein 52 kDa (NDP52) and neighbor of breast cancer susceptibility protein 1 gene 1 (NBR1) were provided by Prof. Michael Lazarou from Monash University and its construction has previously been reported[34].

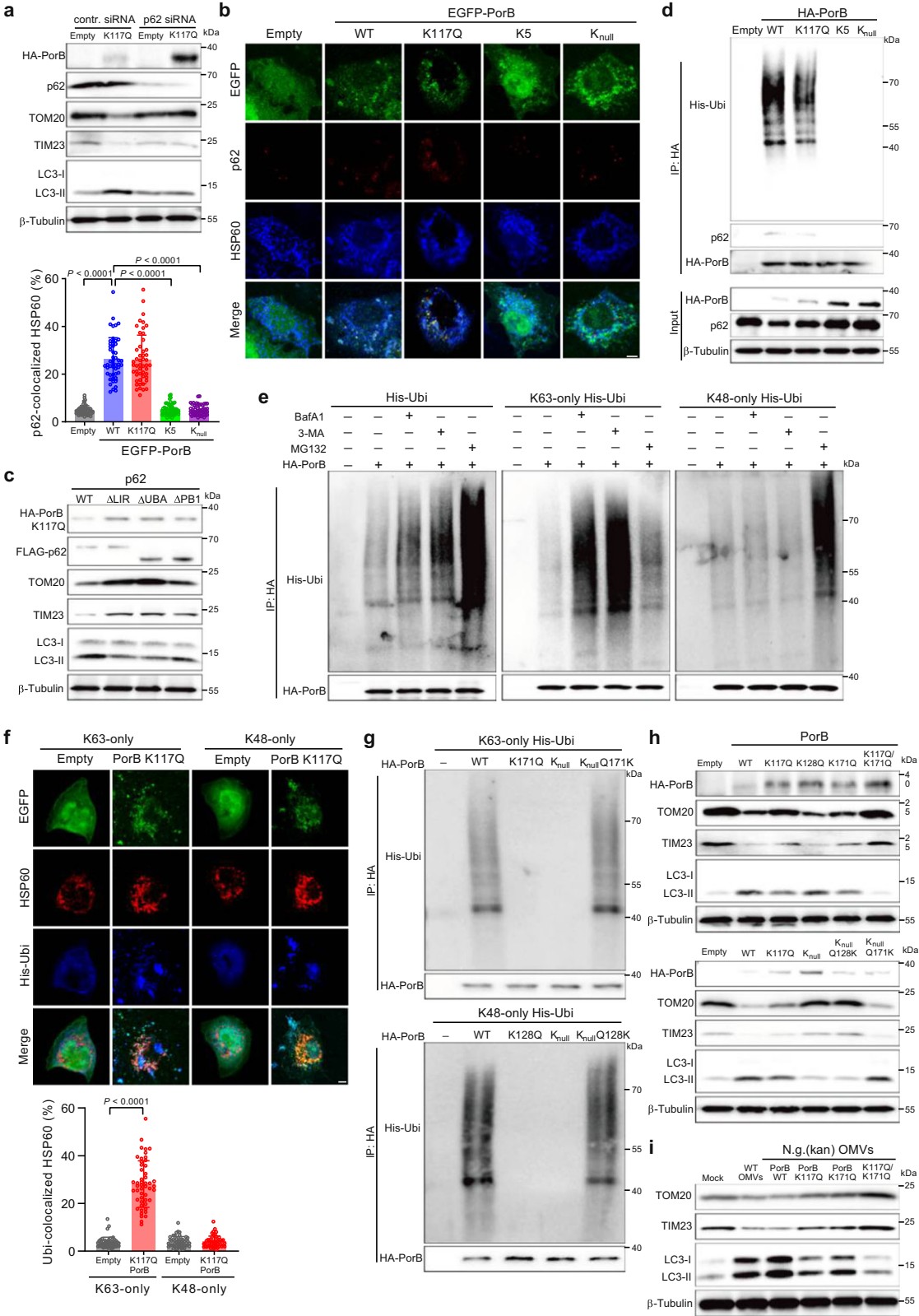

Cell lines were routinely tested for *Mycoplasma* contamination using the Myco-Lumi Luminescent *Mycoplasma* Detection kit (Beyotime, #C0298S) and regularly maintained in MycAway *Mycoplasma* Prevention Reagent (Yeasen, #40608ES03). Lipofectamine 3000 (Thermo Fisher, #L3000075) and optiMEM medium (Gibco, #31985062) were used for transfection of plasmid DNA (2 μg) or siRNA (40 pmol) according to manufacturer's guidelines. Cell were incubated for a

minimum of 24 h before commencement of experiments. For experiments with chemical inhibitors, cells were pretreated with Dynasore (Selleck, #S8047, 80 μM, 1 h), Chlorpromazine (Selleck, #S5749, 15 μg/mL, 1 h), Wortmannin (MCE, #HY-10197, 200 nM, 1 h), Filipin III (Selleck, #HY-N6718, 10 μg/mL, 1 h), BafA1 (Selleck, #S1413, 50 nM, 4 h), 3-MA (10 mM, Selleck, #S2767, 2 h), MG132 (Selleck, #S2619, 5 μM, 6 h), CCCP (Selleck, #S6494, 10 μM, 2 h), Mdivi-1 (Selleck, #S7162, 20 μM, 2 h), or

**Fig. 4 | Lysine 63-linked polyubiquitination of PorB induces p62-dependent mitophagy. a** Western blots showing that knock-down of p62 in HeLa cells inhibits PorB K117Q-dependent degradation of TOM20 and TIM23. **b** Reduced p62 and HSP60 colocalization in HeLa cells expressing PorB K5 or K$_{null}$ compared with PorB WT. Scale bar, 5 μm. Data are mean ± s.d.; $n$ = 50 cells, Kruskal–Wallis with posthoc Dunn test (Empty-WT PorB: $P < 10^{-15}$; WT PorB-PorB K5: $P < 10^{-15}$; WT PorB-PorB K$_{null}$: $P < 10^{-15}$). **c** Western blots of HeLa 4KO cells expressing WT or truncated p62 show that the p62 LIR domain and UBI domain are indispensable for PorB K117Q-induced degradation of TOM20 and TIM23. **d** Western blots showing co-immunoprecipitation of ubiquitin and p62 after immunoprecipitation of PorB WT and PorB K117Q. **e** Western blots showing co-immunoprecipitation of both K63-linked and K48-linked polyubiquitin, after immunoprecipitation of PorB from HeLa cells, which is enhanced with inhibitors for autolysosomal degradation (BafA1, 3-MA) for K63-linked ubiquitin or proteasomal degradation (MG132) for K48-linked ubiquitin. **f** Colocalization between HSP60 and K63-linked polyubiquitin, but not K48-linked polyubiquitin, is induced in HeLa cells expressing PorB. Scale bar, 5 μm. Data are mean ± s.d.; $n$ = 50 cells, two-tailed Mann–Whitney test, K63 only Empty-PorB K117Q: $P < 10^{-15}$. **g** Western blots after immunoprecipitation of HA-PorB from HeLa cells show co-immunoprecipitation of K63-linked polyubiquitin is dependent on PorB lysine 171 and co-immunoprecipitation of K48-linked polyubiquitin is dependent on PorB lysine 128. **h** Western blots showing PorB-induced degradation of TOM20 and TIM23 in HeLa cells is dependent on PorB lysines 117 and 171. **i** Western blots showing gonococcal OMVs expressing PorB K117Q/K171Q lost the ability to induce degradation of TOM20 and TIM23. Cells in **b** and **f** are from 3 independent experiments. Western blots in **a, c, d, e, g, h, i** are representative of 3 independent experiments. Source data are provided as a Source Data file.

Mito-TEMPO (Selleck, #S9733, 10 μM, 2 h). For OMV endocytosis/stimulation assays and gentamicin protections assays, cells were seeded at $2 × 10^5$ cells/well and cultured for 16 h at 37 °C and 5% $CO_2$. Cells were stimulated with OMVs at a dose of 50 μg/well unless otherwise indicated and cells were incubated for 12 h before sample collection unless incubation time was otherwise specified. For OMV cytotoxicity and cell viability assays, HeLa cells were stimulated with OMVs for 24 h before determination of viability and cytotoxicity with the WST-1 Cell Proliferation and Cytotoxicity Assay Kit (Beyotime, C0035) and the LDH Cytotoxicity Assay Kit (Beyotime, C0016). For gentamicin protection assays, cells were challenged with overnight grown bacteria suspended in cell culture media containing 1% (v/v) Vitox at an MOI of 100. Culture plates were centrifuged for 5 min at $800 × g$ and incubated for one hour before sample collection (cell associated). Cells were washed and further incubated with fresh culture medium containing 200 mg/L gentamicin (Sangon Biotech, #B54072). Samples were taken in a time-series (intracellular). For bacterial CFU quantification, cells were lysed with 1% Saponin (Sigma, S10869) and lysates were serially diluted and plated on GC agar containing 1% Vitox. For exocytosis analysis, cells were only incubated for one hour with gentamicin and cell culture medium was subsequently replaced with fresh cell culture medium without gentamicin to allow subsequent quantification of extracellular bacteria in the cell culture medium.

### Expression vectors and siRNA

To silence gene expression, cells were transfected with p62 siRNA (Santa Cruz, #sc-29679), PINK1 siRNA (Santa Cruz, #sc-44598), RNF213 siRNA (Thermo Fisher, 4392420) or the nontargeting Silencer Select negative control No.1 (Thermo Fisher, #4390843). The *N. gonorrhoeae* ATCC 49226 *porB* sequences encoding PorB K$_{null}$ (all lysine residues replaced by glutamine residues), PorB K5 (only retaining the five lysine residues associated with ATP binding in the PorB channel), PorB K$_{strand}$ (only retaining the five surface-exposed lysine residues associated with β-strand motifs and the five lysine residues associated with ATP binding in the PorB channel), PorB K$_{loop}$ (only retaining the four lysine residues associated with surface loops and the five lysine residues associated with ATP binding in the PorB channel) and PorB K$_{peri}$ (only retaining the six lysine residues facing the periplasmic space and the five lysine residues associated with ATP binding in the PorB channel) were synthesized (Tsingke Biotech). For cellular expression of HA-tagged and EGFP-tagged wild-type and mutant PorB from *N. gonorrhoeae* strains ATCC 49226 and ZJXSH86 and from *N. mucosa*, PorB sequences were cloned into vectors pcDNA3.1-HA (Miaoling Bio, #P8326) and pEGFP-C1 (Miaoling Bio, #P0134). Further PorB amino acid mutations (Q48K, Q128K, Q171K, Q269K, K117Q, K128Q, K170Q and K171Q) were generated by mutagenesis PCR of the PorB expression vectors using mutagenesis primers listed in Supplementary Table 1. For all experiments requiring PorB expression, PorB-containing and empty control vectors were transfected to cells. For analysis of autophagy receptor proteins, HeLa 4KO cells were transfected with pcDNA3.1-3×FLAG-OPTN (YouBio, F107017), pcDNA3.1-3×FLAG-NDP52 (YouBio, F101966), pcDNA3.1-3×FLAG-SQSTM1 (YouBio, F101939) or pcDNA3.1-3×FLAG-NBR1, which was generated by cloning NBR1 from pDONR223-NBR1 (YouBio, G114115) to pcDNA3.1-3×FLAG (YouBio, VT8001). For cellular expression of RNF213, HeLa cells were transfected with pcDNA3.1-3×FLAG-RNF213 or pCMV-BFP-RNF213, which were generated by cloning RNF213 from pDONR223-RNF213 (YouBio, G111871) to pcDNA3.1-3×FLAG or pCMV-BFP (Beyotime, D2701). For cellular expression of His-Ubiquitin or His-Ubiquitin that can only generate K63-linked or K48-linked polyubiquitin chains, cells were transfected with pCMV-His-ubiquitin (QCHENG Bio, QCP4836), pCMV-His-ubiquitin(K63) (HedgehogBio, HH-gene-497) or pCMV-His-ubiquitin(K48) (HedgehogBio, HH-gene-496). To investigate contribution of p62 domains, pcDNA3.1-3×FLAG-SQSTM1 was used to generate domain mutant vectors pcDNA3.1-3×FLAG-SQSTM1ΔPB1 (deletion of amino acids 21–85), pcDNA3.1-3×FLAG-SQSTM1ΔLIR (deletion of amino acids 320–340) and pcDNA3.1-3×FLAG-SQSTM1ΔUBA (deletion of amino acids 385–440), which were subsequently transfected to HeLa 4KO cells. To analyze activation of autophagy activity by LC3 puncta, cells were transfected with pEGFP-LC3 (Miaoling Bio, #P0199).

### (Co-)Immunoprecipitation

HeLa cells were cultured in 10 mm dishes (Nest, #704002) and lysed with IP Lysis Buffer (Thermo Fisher, Cat #87787) supplemented with protease inhibitors (Roche, #04693159001). Lysates were centrifuged at $12,000 × g$ for 25 min and supernatant was incubated with balanced anti-HA antibody-coated agarose beads (Lablead, #HNA-25-500) for 2 h at 4 °C. Beads were collected by centrifugation for 2 min at $5000 × g$ and heated for 10 min at 95 °C in SDS-PAGE loading buffer (Beyotime, #P0297) for Western analysis.

### Western analysis

Cells were lysed in RIPA Buffer (Beyotime, #P0013B) containing a protease and phosphatase inhibitor cocktail (NCM biotech, #P002) and samples were heated at 95 °C for 10 min in SDS-PAGE loading buffer (Beyotime, #P0295). Proteins were separated on SDS polyacrylamide gels and transferred onto PVDF membranes (Biorad, #1620177). Membranes were blocked with 5% BSA (Biosharp, #BS114), incubated for 12 h at 4 °C with primary antibodies in primary antibody dilution buffer (Beyotime, #P0023A) and for 2 h at room temperature with HRP-conjugated secondary antibodies in secondary antibody dilution buffer (Beyotime, #P0023D). HRP activity was visualized with Immobilon Western Substrate (Millipore, Cat #WBKLS) and a Chemi-Doc Touch Imaging System (Bio-Rad).

### Immunofluorescence staining and fluorescence microscopy analysis

For fluorescence microscopy procedures, cells were seeded on circular glass coverslips (WHB Scientific, #WHB-12-CS-LC) placed in 12-well

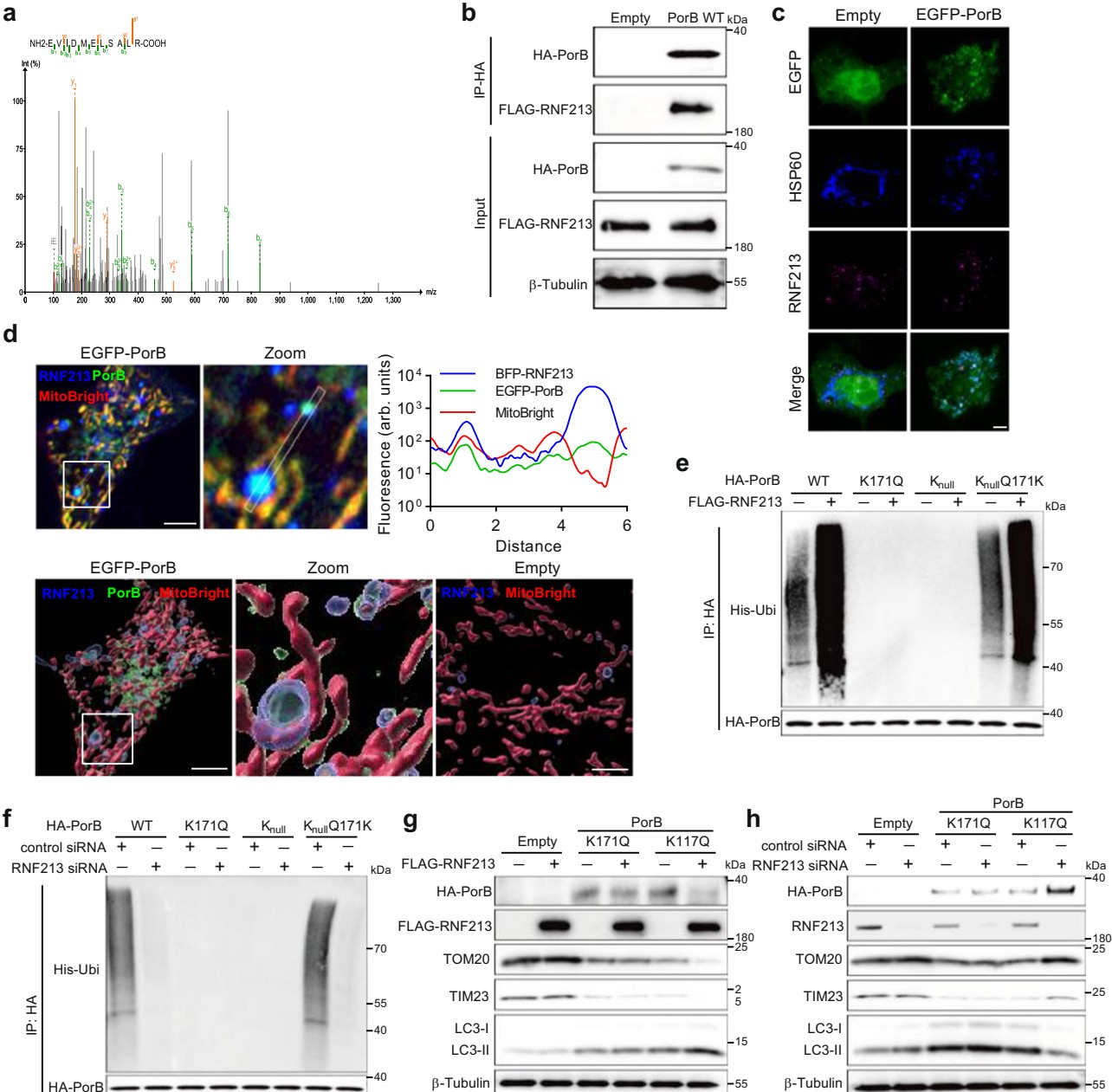

**Fig. 5 | E3 ligase RNF213 interacts with PorB and catalyzes polyubiquitination.**
**a** MS/MS spectrum of RNF213 peptide identified by mass spectrometry analysis of proteins co-immunoprecipitated with PorB from HeLa cells. **b** Western blots after immunoprecipitation of PorB from HeLa cells show co-immunoprecipitation of RNF213. **c** HeLa cells expressing PorB show colocalization between PorB, HSP60 and RNF213. Scale bar, 5 μm. **d** Live HeLa cell microscopy and 3D image reconstruction shows RNF213 surrounding PorB on MitoBright-labeled mitochondria. Scale bar, 5 μm. The fluorescence colocalization profile of the line is shown. **e** Western blots showing PorB K171-dependent enhanced co-immunoprecipitation of ubiquitin from HeLa cells transfected with an RNF213 expression vector.

**f** Western blots showing that transfection of HeLa cells with RNF213 siRNA abolishes PorB K171-dependent co-immunoprecipitation of ubiquitin. **g** Western blots showing PorB K117Q, but not PorB K171Q, enhances PorB-induced degradation of TOM20 and TIM23 in HeLa cells transfected with an RNF213 expression vector. **h** Western blots showing that transfection of HeLa cells with RNF213 siRNA prevents PorB-induced degradation of TOM20 and TIM23 for PorB K117Q, while degradation remains unaffected for PorB K171Q. Western blots in **b**, **e**–**h** are representative of 3 independent experiments. Images in **c** and **d** are representative of 3 independent experiments. Source data are provided as a Source Data file.

plates. Cells, transfected with expression vectors for (fluorescent)-tagged proteins or used in OMV endocytosis or gentamicin protection assays, were washed with PBS and fixed with 4% paraformaldehyde (Biosharp, #BL539A) for 20 min at room temperature. Cells were subsequently incubated with QuickBlock blocking and permeabilization buffer (Beyotime, #P0260) for 40 min before staining for 12 h at 4 °C with primary antibodies diluted in primary antibody dilution buffer (Beyotime, #P0262). Finally, slides were incubated for 2 h at 25 °C with

secondary antibodies diluted in secondary antibody dilution buffer (Beyotime, #P0265). For nuclear staining, 10 μg/mL DAPI solution (Solarbio, #C0065) was used for 15 min at room temperature. Images were acquired on Zeiss LSM880 or Olympus FV3000 microscopes.

**Antibodies**
For immunofluorescence staining or Western analysis, the following antibodies were used: Rabbit anti-LC3A/B (D3U4C) monoclonal

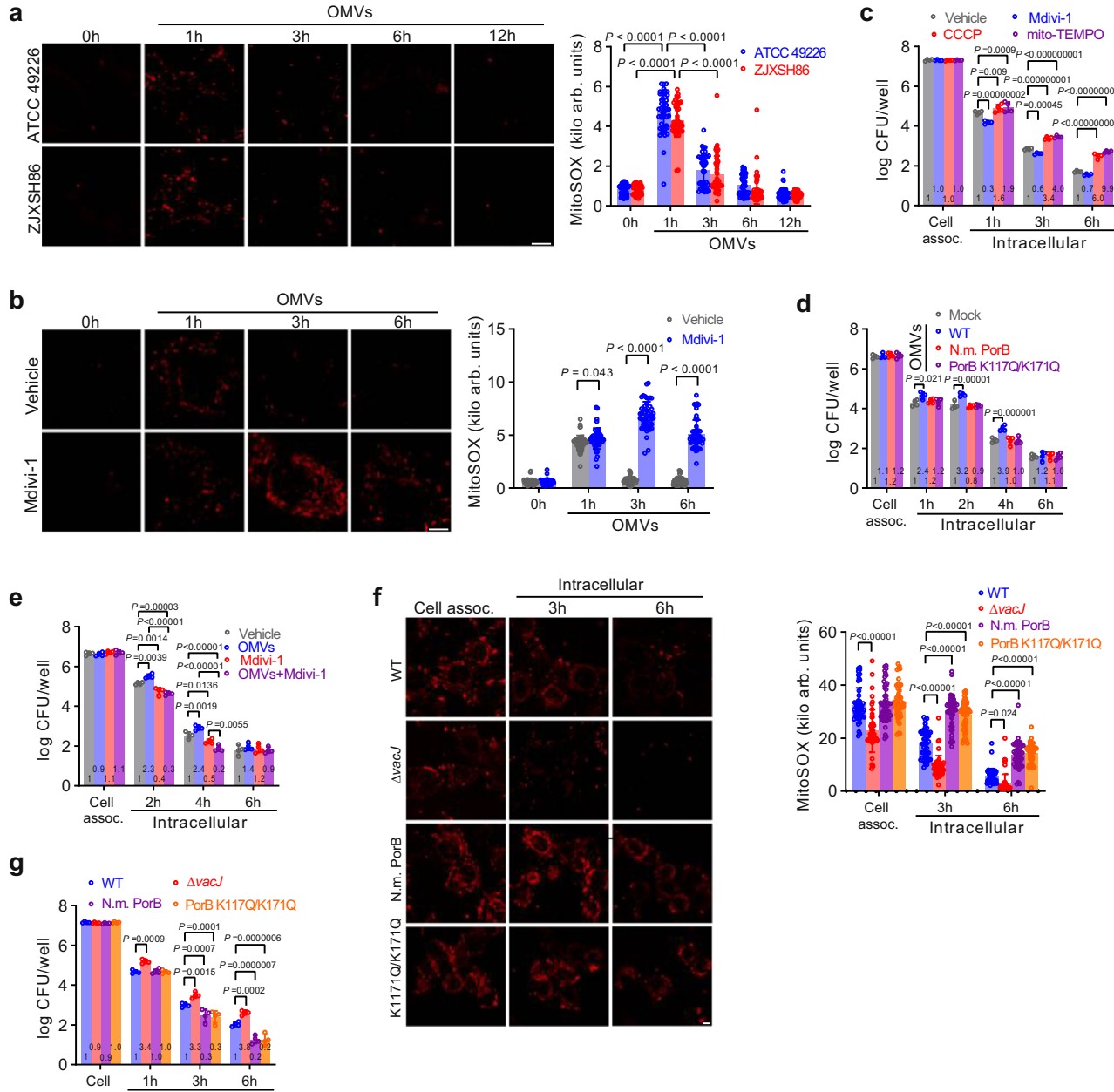

**Fig. 6 | OMV-induced mitophagy reduces generation of mitochondrial ROS to enhance intracellular survival. a** Live-cell microscopy showing that OMVs transiently induce generation of mitochondrial ROS in HeLa cells. Scale bar, 5 µm. Data are mean ± s.d.; $n = 50$ cells, two-way ANOVA with posthoc Bonferroni test, $P < 10^{-15}$ for all reported values. **b** Live-cell microscopy showing that inhibition of mitophagy with Mdivi-1 prolongs mitochondrial generation of ROS in OMV-stimulated HeLa cells. Data are mean ± s.d.; $n = 50$ cells, two-way ANOVA with posthoc Bonferroni test, Vehicle-Mdivi-1: $P < 10^{-15}$ for 3 and 6 h. **c** Inhibition of mitophagy with Mdivi-1 reduces gonococcal intracellular survival in gentamicin protection assays, while activation of mitophagy with CCCP or scavenging of mitochondrial ROS with mito-TEMPO enhances intracellular survival. Data are mean ± s.d.; $n = 4$, two-way ANOVA with posthoc Bonferroni test, Vehicle-mito-TEMPO at 3 h: $P = 4 \times 10^{-11}$, Vehicle-CCCP at 6 h: $P = 1 \times 10^{-14}$, Vehicle-mito-TEMPO at 6 h: $P < 10^{-15}$. **d** Prior stimulation of HeLa cells with gonococcal WT OMVs to induce mitophagy enhances gonococcal intracellular survival in gentamicin protection assays, while OMVs from gonococcal mutants expressing PorB from *N. mucosa* or PorB K117Q/K171Q are unable to enhance intracellular survival. Data are mean ± s.d.; $n = 4$, two-way ANOVA with posthoc Bonferroni test. **e** Prior stimulation of Mdivi-1-pretreated HeLa cells with

gonococcal OMVs reduces gonococcal intracellular survival in gentamicin protection assays. Data are mean ± s.d.; $n = 4$, two-way ANOVA with posthoc Bonferroni test, OMVs-OMVs+Mdivi-1 at 2 h: $P = 9 \times 10^{-11}$, Vehicle-OMVs+Mdivi1 at 4 h: $P = 2 \times 10^{-7}$, OMVs-OMVs+Mdivi-1 at 4 h: $P < 10^{-15}$. **f** Live-cell microscopy of gonococcal-challenged HeLa cells showing reduced generation of mitochondrial ROS for the Δ*vacJ* mutant and enhanced generation of mitochondrial ROS for *N. gonorrhoeae* expressing PorB from *N. mucosa* or PorB K117Q/K171Q. Data are mean ± s.d.; $n = 50$ cells, two-way ANOVA with posthoc Bonferroni test, WT- Δ*vacJ* at cell associated: $P < 10^{-15}$, WT-Δ*vacJ* at 3 h: $P = 1 \times 10^{-13}$, WT-N.m. PorB at 3 h: $P < 10^{-15}$, WT-PorB K117Q/K171Q at 3 h: $P < 10^{-15}$, WT-N.m. PorB at 6 h: $P = 4 \times 10^{-12}$, WT-PorB K117Q/K171Q at 6 h: $P = 6 \times 10^{-14}$. **g** Gonococcal mutants expressing PorB from *N. mucosa* or PorB K117Q/K171Q show reduced intracellular survival in gentamicin protection assays. Data are mean ± s.d.; $n = 4$, two-way ANOVA with posthoc Bonferroni test. Cells in **a, b, f** are from 3 independent experiments. Data in **c–e, g** are log-normalized CFU per well from 4 independent experiments, with fold-changes in survival compared with the Vehicle or Mock provided within the bars. Source data are provided as a Source Data file.

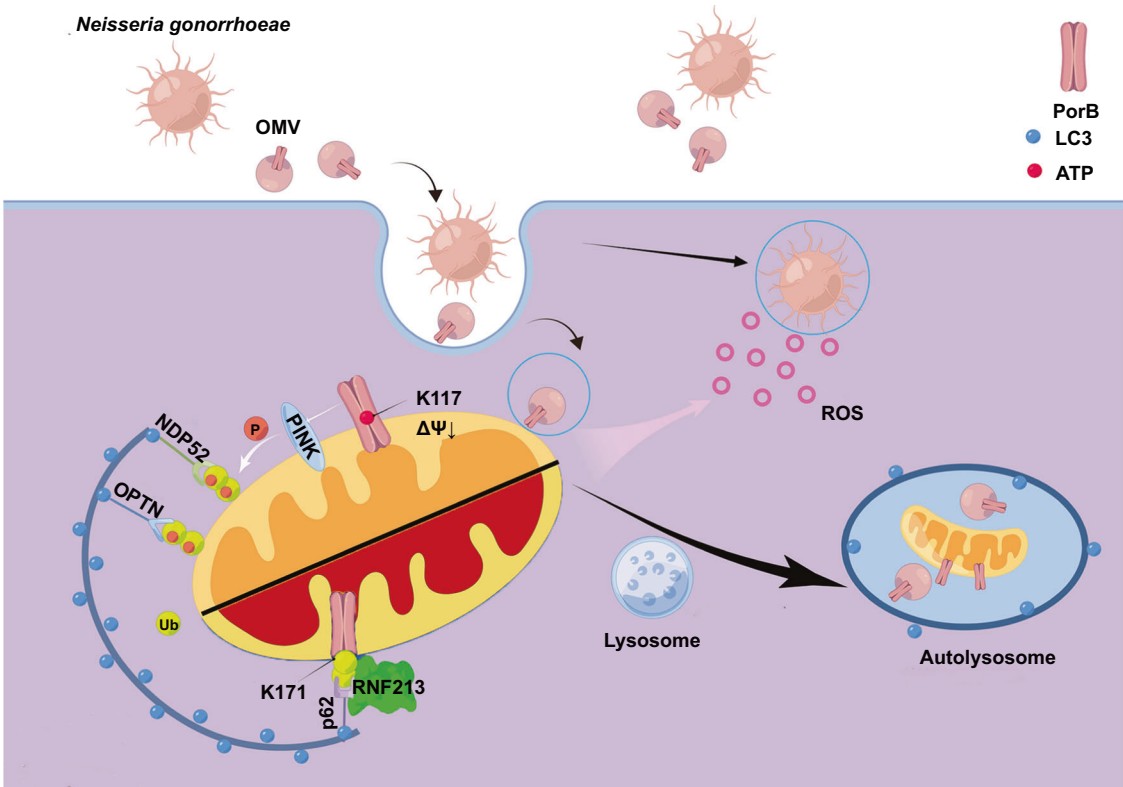

**Fig. 7 | Summary illustration of the gonococcal OMV-induced mitophagy mechanism through a dual PorB-dependent pathway.** Gonococcal OMVs are endocytosed by epithelial cells and deliver PorB to the mitochondria. Insertion of PorB in the mitochondrial membrane dissipates the mitochondrial membrane potential (MMP). Dissipation of MMP is dependent on ATP binding by lysines in the PorB channel, such as lysine 117, that prolong the channel in an open position. Dissipation of MMP results in PINK1- and OPTN/NDP52-dependent activation of mitophagy. PorB is furthermore decorated by K63-linked polyubiquitin chains at PorB lysine 171 through activity of E3 ubiquitin ligase RNF213, which activates mitophagy in a p62-dependent manner. OMV- and PorB-dependent activation of mitophagy abolishes mitochondrial generation of reactive oxygen species (ROS) upon gonococcal invasion of epithelial cells, which results in enhanced intracellular survival. Figure created with cartoon components by Figdraw [www.figdraw.com].

antibodies (CST, #12741, Western 1:1000, immunofluorescence 1:200); Rabbit anti-LAMP1 (D2D11) monoclonal antibodies (CST, #9091, immunofluorescence 1:200); Rabbit anti-HA (C29F4) monoclonal antibodies (CST, #3724, Western 1:1000); Rabbit anti-His (D3I1O) monoclonal antibodies (CST, #12698, Western 1:1000, immunofluorescence 1:200); Rabbit anti-TOM20 (D8T4N) monoclonal antibodies (CST, #42406, Western 1:1000, immunofluorescence 1:200); Rabbit anti-p62 (D6M5X) monoclonal antibodies (CST, #23214, immunofluorescence 1:200); Mouse anti-FLAG (M2) monoclonal antibodies (Sigma, #F1804, Western 1:1000); Rabbit anti-TIM23 polyclonal antibodies (Proteintech, #11123-1-AP, Western 1:1000); Rabbit anti-p62 polyclonal antibodies (Proteintech, #18420-1-AP, Western 1:1000); Rabbit anti-p62 (PS00-61) monoclonal antibodies (Huabio, #HA721171, Western 1:1000); Rabbit anti-PINK1 polyclonal antibodies (Proteintech, #23274-1-AP, Western 1:500); Mouse anti-β-Tubulin polyclonal antibodies (Proteintech, #10068-1-AP, Western 1:1000); Mouse anti-HSP60 (H-1) monoclonal antibodies (Santa Cruz, #sc-13115, immunofluorescence 1:200); Rabbit anti-RNF213 polyclonal antibodies (Merck, #HPA003347, immunofluorescence 1:100, Western 1:500); Rabbit anti-PDI polyclonal antibodies (Proteintec, #11245-1-AP, Western 1:500); Rabbit anti-GM130 polyclonal antibodies (Beyotime, #AF7005, Western 1:500); HRP-conjugated goat anti-rabbit IgG (H + L) secondary antibodies (Beyotime, #A0208, Western 1:1000); HRP-conjugated goat anti-mouse IgG (H + L) secondary antibodies (Beyotime, #A0216, Western 1:1000); Goat anti-rabbit IgG (H + L) Alexa Fluor 488 cross-adsorbed secondary antibodies (Thermo Fisher, #A-11008,

immunofluorescence 1:200); Goat anti-mouse IgG (H + L) Alexa Fluor 405 cross-adsorbed secondary antibodies (Thermo Fisher, #A-31553, immunofluorescence 1:200); Goat anti-rabbit IgG (H + L) Alexa Fluor 594 cross-adsorbed secondary antibodies (Thermo Fisher, #A-11012, immunofluorescence 1:200); Goat anti-mouse IgG (H + L) Alexa Fluor 594 cross-adsorbed secondary antibodies (Thermo Fisher, #A-11032, immunofluorescence 1:200).

### Live cell imaging for MMP, ROS and high-resolution microscopy analysis

HeLa cells ($2 \times 10^5$ cells/dish) were seeded in glass bottom cell culture dishes (NEST, #801002) and transfected with expression vectors for (fluorescent)-tagged proteins or used in OMV endocytosis or gentamicin protection assays. For identification of endosomes, Cascade Blue-conjugated dextran 10,000 MW (Thermo Fisher, #D1976) was added to cell culture media (5 mg/mL). For specific fluorescent labeling of mitochondria, MitoBright LT Red (DOJINDO, #MT11) was added to the cell culture medium (0.1 μM). For evaluation of the mitochondrial membrane potential, TMRM (Thermo Fisher, #T668) was added to the cell culture medium (100 mM). For detection of mitochondrial ROS, MitoSOX (Yeasen, #40778ES50) was added to the cell culture medium (5 μM). Images were acquired on Zeiss LSM880 or Olympus FV3000 microscopes while maintaining the cells in a heated $CO_2$ incubator. High-resolution images acquired with the Zeiss LSM880 microscope were used for 3D reconstruction with Imaris 9.5 software.

## Transmission electron microscopy

Cells and bacteria or OMVs were fixed overnight with 2.5% glutaraldehyde and incubated for 30 min in 2% osmium tetroxide in 0.1 M sodium cacodylate buffer. After dehydration in graded alcohols, samples were embedded in 812 epoxy resin (Electron Microscopy Sciences). Ultramicrotome-cut sections were stained with uranyl acetate and lead citrate and visualized with a Tecnai G2 Spirit transmission electron microscope (FEI Company).

## Quantitative real-time PCR

DNA was extracted with the TIANamp genomic DNA extraction kit (Tiangen, #DP304) following manufacturer's instructions. Quantitative real-time PCR reactions were performed in a total volume of 20 μL using 10 μL SYBR Green Master Mix (Vazyme, #Q111-02), 2 μL DNA and 200 nM of COXII (mitochondrial DNA) or RPL13A (genomic DNA) primers (Supplementary Table 1). Reactions were run on a Roche 480II with an initial step of 30 s at 95 °C, followed by 40 cycles of 95 °C for 10 s and 60 °C for 30 s. Relative differences between mitochondrial and genomic DNA were calculated by the $2^{-\Delta\Delta Ct}$ method.

## Flow cytometry

Cells were washed and dissociated using trypsin without EDTA (Beyotime, #C0207). Induction of apoptosis was determined with the Annexin V, FITC Apoptosis Detection Kit (DOJINDO, #AD10) according to manufacturer's instructions. Loss of mitochondrial membrane potential was evaluated with TMRM (100 mM, Thermo Fisher, #T668). Mitochondrial ROS were detected with MitoSOX (5 μM, Yeasen, #40778ES50). Endocytosis of DiO-labeled OMVs after pre-treatment of cells with chemical inhibitors was determined 2 h after OMV challenge. All samples were run on a DxFLEX (Beckman) following the gating strategy in Supplementary Fig. 8 to select for single cells and Flow Jo X 10.0.7 software was used for data analysis.

## LC-MS/MS analysis

To identify possible E3 ubiquitin ligases involved in PorB ubiquitination, vectors expressing HA-PorB and His-Ubiquitin were co-transfected to HeLa cells and PorB was immunoprecipitated with anti-HA antibody-coated agarose beads (Lablead, #HNA-25-500). Beads were collected and heated for 10 min at 95 °C in SDS-PAGE loading buffer (Beyotime, #P0297) and subsequently send to the Beijing Genomics Institute (BGI) for further processing and identification of co-immunoprecipitated proteins by LC-MS/MS. Shortly, the sample was compressed by SDS-PAGE, dehydrated with acetonitrile, and Trypsin digested. Peptide fragments were separated on an UltiMate 3000 UHPLC (Thermo Fisher) and analyzed by a Q-Exactive HF tandem mass spectrometer (Thermo Fisher). Original MS data were loaded into Mascot 2.3.02 software and the UniProt database [https://www.uniprot.org/] was used for identification of human protein sequences. Percolator was used to improve and correct matching proteins. Mass spectrometry data have been deposited to the iProX repository (https://www.iprox.cn/) with identifier IPX0007863000.

## Quantification and statistics

Quantification of fluorescent signal colocalization in images was determined with the Fiji ImageJ colocalization plugin, and a positive pixel was defined at a minimum intensity of 50 out of the maximum intensity of 255. Puncta were defined by clusters of at least 50 positive pixels. Quantitative microscopy analyses included at least 50 cells for each group. Results were plotted and analyzed using GraphPad Prism 8.0.2 software. Quantitative data was presented as mean ± s.d. from a minimum of three biological replicates. Normal distribution of datasets was assessed with the Shapiro–Wilk test. For comparisons between two groups, the unpaired two-tailed $t$-test or two-tailed Mann–Whitney test were used, while for comparisons between multiple groups one-way ANOVA with posthoc Tukey test, two-way ANOVA with posthoc Bonferroni test or Kruskal–Wallis with posthoc Dunn test were used. A $P < 0.05$ was considered significantly different.

## Reporting summary

Further information on research design is available in the Nature Portfolio Reporting Summary linked to this article.

## Data availability

Mass spectrometry data generated in this study have been deposited in the iProX database under accession code IPX0007863000. Uniprot [https://www.uniprot.org/] was used for identification of human proteins. All source data generated in this study are provided in the Source Data/Supplementary Information file. This also includes blots. Materials and associated protocols are available upon request. Source data are provided with this paper.

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

## Acknowledgements

This work was supported by the National Natural Science Foundation of China (grant numbers 82272382, 82150610507, 82072320, 81871695) and the Zhejiang Province Natural Science Foundation (grant number Z24H190002). The funder had no role in study design, data collection and interpretation, writing of the manuscript, or the decision to submit the manuscript for publication. We thank Lin Zhaoxiaonan and Xiao Guifeng from the Core Facilities of Zhejiang University School of Medicine for assistance with confocal microscopy. We thank Prof Michael Lazarou from Monash University for generously providing the HeLa quadruple KO cell line for autophagy receptors.

## Author contributions

S.G. and L.G. are joined-first author. S.V. conceived the project. S.G., L.G. and S.V. designed the experiments. S.G., L.G. and D.Y. performed the experiments. S.G., L.G., X.L., and S.V. analyzed the data. S.G. and S.V. wrote the manuscript. X.L. edited the manuscript. All author approved the final manuscript.

## Competing interests

The authors declare no competing interests.
