## [Peer Review File · Nature Communications]

REVIEWER COMMENTS

Reviewer #1 (Remarks to the Author):

The current manuscript is focused on whether PorB containing OMVs induce mitophagy in epithelial cells and whether this promotes *Neisseria gonorrhoeae* infections. The manuscript is well presented and contains several important observations. The experiments are clearly described and several methods/approaches have been applied to support the major findings. The new findings reported here will be of great interest to the host-pathogen interaction community. I do not have any major concerns and support publication of the manuscript. The below points are merely for consideration to increase clarity.

1. Figure 1b: label or figure legend indicating the meaning of the numbers is missing

2. Figure 1: LC3 also contributes to phagocytosis but this has not been addressed. Additional autophagy markers, such as loss of p62, should be included to support upregulation of autophagy.

3. Line 86 “Endosomal delivery of OMVs to mitochondria”: not clear what this means. OMVs traffic to endosomes. Are you suggesting that endosomes directly interact with mitochondria? Any evidence that OMVs traffic to lysosomes or the cytosol?

Does endosome tracker dextran cascade escape endosomes together with OMVs and target mitochondria?

4. Figure 6c, d f: Consider including fold-changes in the figure or main text. Some of the differences seem marginal but this is likely due to the scaling.

5. Given that OMV/m-divi cause substantial increased MitoSOX staining, have you considered testing whether the combination affects intracellular CFUs even more so other conditions?

Reviewer #2 (Remarks to the Author):

The paper by Gao et al. investigates the role of outer membrane vesicles (OMVs) of pathogenic bacteria (*N. gonorrhoeae*) in the intracellular survival of the pathogen within endothelial cells. The authors show that PorB present within the OMVs is sufficient to trigger mitophagy and reduce the formation of ROS in HeLa cells. Two pathways are shown to be implicated in the induction of mitophagy by gonococcal OMVs. The first pathway is dependent on PorB mediated loss of mitochondrial membrane potential, which triggers the classical PINK1 mechanism involving OPTN/NDP52 recruitment. A second pathway shown in the paper triggers RNF213-dependent K63-linked polyubiquitination on PorB residue K171 and p62 recruitment. Overall the data are of good quality and the methodology is sound, although often the number of replicates is not indicated.

Can the authors clarify how many replicates were performed for the imaging and for the Western blot experiments? For imaging, the number of cells is stated, but not whether these cells were from a single or from multiple replicated experiments.

For the TEM images in Figures 1a and 3a can the authors add inserts with zoom ins similar to Figure 2e? Also, can the authors provide TEM images of the purified OMVs used in subsequent experiments?

Can the authors discuss why PorB from *N.mucosa* does not target to mitochondria? Would targeting of the *N.mucosa* PorB to mitochondria result in mitophagy induction? Are the same motifs for K63-linked polyubiquitination and for ATP binding present in *N.m.* PorB?

Can the authors comment on why the overall effects of mitophagy inhibition or pre stimulation with OMVs on intracellular survival are rather subtle in Figure 6?

The scheme in Extended Data Figure 8 could present a clearer picture of the two distinct pathways that induce mitophagy.

Reviewer #1 (Remarks to the Author):

THE CURRENT MANUSCRIPT IS FOCUSED ON WHETHER PORB CONTAINING OMVS INDUCE MITOPHAGY IN EPITHELIAL CELLS AND WHETHER THIS PROMOTES NEISSERIA GONORRHOEAE INFECTIONS. THE MANUSCRIPT IS WELL PRESENTED AND CONTAINS SEVERAL IMPORTANT OBSERVATIONS. THE EXPERIMENTS ARE CLEARLY DESCRIBED AND SEVERAL METHODS/APPROACHES HAVE BEEN APPLIED TO SUPPORT THE MAJOR FINDINGS. THE NEW FINDINGS REPORTED HERE WILL BE OF GREAT INTEREST TO THE HOST-PATHOGEN INTERACTION COMMUNITY. I DO NOT HAVE ANY MAJOR CONCERNS AND SUPPORT PUBLICATION OF THE MANUSCRIPT. THE BELOW POINTS ARE MERELY FOR CONSIDERATION TO INCREASE CLARITY.

We thank the reviewer for a thorough review of our manuscript and the positive evaluation. We have addressed all questions and comments raised by the reviewer and improved our manuscript accordingly.

1. FIGURE 1B: LABEL OR FIGURE LEGEND INDICATING THE MEANING OF THE NUMBERS IS MISSING

As suggested by the reviewer, we improved labelling of Figure 1b and provided additional explanations in the figure legend to describe more clearly what the graph represents. We now also included in the graph the relative differences in intracellular survival and exocytosis between the wild-type strain and $\Delta vacJ$ mutant to highlight the differences more clearly, since differences on a log-scale graph might appear minimal. Specifically, the graph now states on the y-axis “Log CFU/well” and the description in the figure legend states “Enhanced intracellular survival and exocytosis of gonococcal $\Delta vacJ$ mutant in gentamicin protection assays with HeLa cells. Data are mean \pm s.d. of log-normalized colony forming units (CFU) per well and relative differences in survival between WT and $\Delta vacJ$ are provided within the bars”.

2. FIGURE 1: LC3 ALSO CONTRIBUTES TO PHAGOCYTOSIS BUT THIS HAS NOT BEEN ADDRESSED. ADDITIONAL AUTOPHAGY MARKERS, SUCH AS LOSS OF P62, SHOULD BE INCLUDED TO SUPPORT OF UPREGULATION OF AUTOPHAGY.

We agree with the reviewer that LC3 could also contribute to autophagy-independent phagocytosis/endocytosis processes. Therefore, as suggested by the reviewer, we now included additional Western blots of the autophagy marker protein p62 in Figures 1e and 1h. These Western blots show reduced p62 levels after OMV endocytosis, while pre-treatment with BafA1 resulted in increased p62 levels. These additional results are

fully in support of the LC3 Western blots and demonstrate activation of autophagy after OMV endocytosis.

3. LINE 86 “ENDOSOMAL DELIVERY OF OMVS TO MITOCHONDRIA”: NOT CLEAR WHAT THIS MEANS. OMVS TRAFFIC TO ENDOSOMES. ARE YOU SUGGESTING THAT ENDOSOMES DIRECTLY INTERACT WITH MITOCHONDRIA? ANY EVIDENCE THAT OMVS TRAFFIC TO LYSOSOMES OR THE CYTOSOL?

DOES ENDOSOME TRACKER DEXTRAN CASCADE ESCAPE ENDOSOMES TOGETHER WITH OMVS AND TARGET MITOCHONDRIA?

We agree with the reviewer that the phrase “endosomal delivery of OMVs to the mitochondria” is not very clear. The OMVs are endocytosed by epithelial cells and therefore present within endosomes during subsequent intracellular trafficking. The endosomes containing the OMVs appear to accumulate strongly around mitochondria, as demonstrated by our high-resolution microscopy experiments (Fig. 2c). To state this more clearly, we modified the sentence to: “Indeed, OMVs surrounding or engaging with mitochondria were readily observable (Fig. 2b), while high-resolution confocal microscopy with the endosome tracker dextran cascade blue showed mitochondria surrounded by OMVs that are predominantly present within endosomes (Fig. 2c).”

Based on our experiments, it is not possible to determine conclusively about direct physical interactions between OMV-containing endosomes and mitochondria, however, based on their close vicinity, direct contact is possible. Our preliminary OMV trafficking analyses indeed demonstrated OMV colocalization with lysosomes during later time points, but cytosol localization of OMVs should be minimal since we did not detect abundant OMV signals that are not colocalized with endosome tracker dextran cascade blue. Based on 3D reconstruction of high-resolution microscopy images, it is not possible to detect the endosome tracker dextran cascade blue when present in the cytosol, since the signal will be lost due to diffusion. We now included a section in the Discussion regarding OMV/endosome trafficking and possible interactions with mitochondria in relation with PorB translocation.

4. FIGURE 6C, D, F: CONSIDER INCLUDING FOLD-CHANGES IN THE FIGURE OR MAIN TEXT. SOME OF THE DIFFERENCES SEEM MARGINAL BUT THIS IS LIKELY DUE TO THE SCALING.

Indeed, the graphs presented in these figure panels are log-scale, resulting in the appearance of small differences. Log-scale is commonly used for bacterial survival data and because the differences between the different time-points are large due to a combination of intracellular killing and exocytosis of surviving bacteria, we have to use log scale to visualize all time-points in the same graph. Therefore, as suggested by the reviewer, we included fold-changes in the graph to highlight differences between the specific conditions compared with the Vehicle or Mock controls are for some

conditions up to 9.9-fold (Fig. 6c), 3.9-fold (Fig. 6d), 2.4-fold (Fig. 1e) and 3.8-fold (Fig. 1g). We included the statement “Data in **c**, **d**, **e**, **g** are log-normalized CFU per well from 4 independent experiments, with fold-changes in survival compared with the Vehicle or Mock provided within the bars” in the figure legend regarding the inclusion of fold changes.

5. GIVEN THAT OMV/M-DIVI CAUSE SUBSTANTIAL INCREASED MITOSOX STAINING, HAVE YOU CONSIDERED TESTING WHETHER THE COMBINATION AFFECTS INTRACELLULAR CFUS EVEN MORE SO OTHER CONDITIONS?

As suggested by the reviewer, we now also tested intracellular survival after OMV stimulation of mDivi-1 pre-treated cells. The OMV/mDivi-1 combination resulted in significantly reduced intracellular survival compared with the control condition or OMV stimulation condition. Furthermore, at the 4h time point the OMV/mDivi-1 condition also showed significantly reduced intracellular survival compared with the mDivi-1 only condition, supporting that increased MitoSOX staining/ROS formation reduces intracellular survival. These new results have been included as Figure 6e and subsequent panels have been renumbered.

Reviewer #2 (Remarks to the Author):

THE PAPER BY GAO ET AL. INVESTIGATES THE ROLE OF OUTER MEMBRANE VESICLES (OMVS) OF PATHOGENIC BACTERIA (N. GONORRHOEAE) IN THE INTRACELLULAR SURVIVAL OF THE PATHOGEN WITHIN ENDOTHELIAL CELLS. THE AUTHORS SHOW THAT PORB PRESENT WITHIN THE OMVS IS SUFFICIENT TO TRIGGER MITOPHAGY AND REDUCE THE FORMATION OF ROS IN HELA CELLS. TWO PATHWAYS ARE SHOWN TO BE IMPLICATED IN THE INDUCTION OF MITOPHAGY BY GONOCOCCAL OMVS. THE FIRST PATHWAY IS DEPENDENT ON PORB MEDIATED LOSS OF MITOCHONDRIAL MEMBRANE POTENTIAL, WHICH TRIGGERS THE CLASSICAL PINK1 MECHANISM INVOLVING OPTN/NDP52 RECRUITMENT. A SECOND PATHWAY SHOWN IN THE PAPER TRIGGERS RNF213-DEPENDENT K63-LINKED POLYUBIQUITINATION ON PORB RESIDUE K171 AND P62 RECRUITMENT. OVERALL THE DATA ARE OF GOOD QUALITY AND THE METHODOLOGY IS SOUND, ALTHOUGH OFTEN THE NUMBER OF REPLICATES IS NOT INDICATED.

We thank the reviewer for a thorough and positive evaluation of our manuscript. We have improved the manuscript according to the reviewer’s suggestions and specifically included the number of replicates of all quantitative data and Western blots in the figure legends.

CAN THE AUTHORS CLARIFY HOW MANY REPLICATES WERE PERFORMED FOR THE IMAGING AND FOR THE WESTERN BLOT EXPERIMENTS? FOR IMAGING, THE NUMBER OF CELLS IS STATED, BUT NOT WHETHER THESE CELLS WERE FROM A SINGLE OR FROM MULTIPLE REPLICATED EXPERIMENTS.

As suggested by the reviewer, we now included statements in the figure legends indicating the number of replicates for Western blotting and imaging. Specifically, Western blots are representative of three independent experiments, while for imaging we already included the number of cells analysed and additionally included the number of independent experiments these cells represent.

FOR THE TEM IMAGES IN FIGURES 1A AND 3A CAN THE AUTHORS ADD INSERTS WITH ZOOM INS SIMILAR TO FIGURE 2E? ALSO, CAN THE AUTHORS PROVIDE TEM IMAGES OF THE PURIFIED OMVs USED IN SUBSEQUENT EXPERIMENTS?

As suggested by the reviewer, we now included zoom ins for the TEM images in Figures 1a and 3a. For Figure 2e we included these zoom ins as separate images below the original images. However, due to figure size constraints and organization of the figure panels, we included the zoom ins for Figures 1a and 3a as partial overlays of the original images. We also included additional zoom ins for the TEM images presented in Supplementary Figure 1e. Furthermore, we now also included additional TEM images of the purified OMVs from *N. gonorrhoeae* strains ATCC 49226 and ZJXSH86 used in further OMV experiments. These images have been included as Figure 1d and subsequent panels of Figure 1 have been renumbered accordingly.

CAN THE AUTHORS DISCUSS WHY PORB FROM N. MUCOSA DOES NOT TARGET TO MITOCHONDRIA? WOULD TARGETING OF THE N. MUCOSA PORB TO MITOCHONDRIA RESULT IN MITOPHAGY INDUCTION? ARE THE SAME MOTIFS FOR K63-LINKED POLYUBIQUITINATION AND FOR ATP BINDING PRESENT IN N.M. PORB?

As suggested by the reviewer, we included an additional paragraph in the Discussion section regarding PorB translocation to mitochondria and comparing gonococcal PorB and PorB from *Neisseria mucosa*. It is currently not fully elucidated how gonococcal PorB targets mitochondria, however, it has been demonstrated that the C-terminal quarter of gonococcal PorB is important for mitochondrial import and sufficient to target other proteins to mitochondria. However, other PorB domains also appear to contribute to mitochondrial import. For mitochondrial β -barrel proteins it has been demonstrated that the hydrophobicity of cytoplasm-facing amino acids in the β -hairpin motifs of these proteins determines mitochondrial targeting, with the final C-terminal hairpin displaying a dominant, but not exclusive, role. However, the β -hairpin motifs of gonococcal PorB are not as hydrophobic as the motifs from mitochondrial β -barrel

proteins such as human VDAC1. Furthermore, the hydrophobicity of cytoplasm-facing amino acids in the C-terminal quarter β -hairpins of gonococcal PorB and PorB from *N. mucosa* do not differ noteworthy and can therefore not explain the inability of *N. mucosa* PorB to target mitochondria. PorB from *N. mucosa* does contain the equivalent lysine residue at position 117, however, it has previously been demonstrated that PorB from *N. mucosa* does not bind ATP, likely because it lacks two of the other lysine residues in the barrel channel that directly interact with ATP. Furthermore, gonococcal PorB lysine residue 171, which is decorated with K63-linked polyubiquitin, is not present in the PorB from *N. mucosa*. However, a lysine residue is present in another position at the equivalent much larger loop of *N. mucosa*. Therefore, we cannot fully exclude possible K63-linked polyubiquitination of this lysine residue.

CAN THE AUTHORS COMMENT ON WHY THE OVERALL EFFECTS OF MITOPHAGY INHIBITION OR PRE STIMULATION WITH OMVS ON INTRACELLULAR SURVIVAL ARE RATHER SUBTLE IN FIGURE 6?

Actually, the effects of mitophagy inhibition (down to 0.3-fold) and OMV pre-stimulation (up to 3.2-fold) on intracellular survival are not very subtle, however, because the graphs presented in Figure 6 are log-scale, the differences between the mitophagy-inhibiting or mitophagy-inducing conditions compared with the Vehicle or Mock controls appear limited. Therefore, as suggested by Reviewer 1 in comment 4, we included fold-changes in these graphs to highlight the magnitude of the differences in intracellular survival between the specific conditions and the Vehicle or Mock controls. We included statements regarding the inclusion of fold-changes in the figure legends of the specific panels of Figure 6.

THE SCHEME IN EXTENDED DATA FIGURE 8 COULD PRESENT A CLEARER PICTURE OF THE TWO DISTINCT PATHWAYS THAT INDUCE MITOPHAGY.

As suggested by the reviewer, we have modified the scheme to present more clearly the two separate PorB-dependent pathways able to induce mitophagy. Specifically, we have colour-divided the mitochondrion in two distinct colours for top and bottom, with each part presenting one of the two identified mechanisms. In our revised manuscript, we have included this scheme as Figure 7.

REVIEWERS' COMMENTS

Reviewer #1 (Remarks to the Author):

The authors have adequately addressed my previous concerns.